# Building Bricks of Integrated Care Pathway for Autism Spectrum Disorder: A Systematic Review

**DOI:** 10.3390/ijms24076222

**Published:** 2023-03-26

**Authors:** Francesca Fulceri, Letizia Gila, Angela Caruso, Martina Micai, Giovanna Romano, Maria Luisa Scattoni

**Affiliations:** 1Istituto Superiore di Sanità, Research Coordination and Support Service, 00161 Rome, Italy; 2Directorate General of Health Prevention, Ministry of Health, 00144 Rome, Italy

**Keywords:** care pathway, autism spectrum disorder, services, guidelines, care coordination

## Abstract

An integrated plan within a defined care pathway for the diagnosis, continuative interventions, and periodic redefinition of care of autistic people is essential for better outcomes. Challenges include delivering services across all domains or life stages and effective coordination between health/social care providers and services. Further, in the ‘real world’, service provision varies greatly, and in many settings is significantly weighted towards diagnosis and children’s services rather than treatment and support or adult care. This study aims to identify existing care pathways for Autism Spectrum Disorder (ASD) from referral to care management after diagnosis. The study reviewed the international literature in PubMed and PsycInfo databases and collected information on care for autistic individuals from the Autism Spectrum Disorders in Europe (ASDEU) project partners. The study found that published data mainly focused on specific components of care pathways rather than an integrated and coordinated plan of care and legislative indications. They should be aimed at facilitating access to the services for support and the inclusiveness of autistic individuals. Given the need for care addressing the complex and heterogeneous nature of ASD, effective coordination between different health/social care providers and services is essential. It is also suggested that research priority should be given to the identification of an integrated care pathway ‘model’ centered around case management, individualization, facilitation, support, continuous training and updating, and quality management.

## 1. Introduction

Autism—or autism spectrum disorder (ASD), the formal term used in the *American Psychiatric Association’s Diagnostic and Statistical Manual of Mental Disorders, 5th edition (DSM-5)*—is a heterogeneous disorder characterized by persistent deficits in social communication and social interaction in various contexts and restricted and/or repetitive patterns of behavior, interests, and activities [1]. Clinical symptoms vary across life, and often autistic individuals present more than one neurological, psychiatric, and medical comorbidity [2]. A single assessment or a single treatment is never sufficient [3], and services that autistic individuals and their families need to rely on evolve across different stages of their life. Children require early intervention, behavior management, and school support [4,5,6], while adolescents need services related to community programs, school success, and life skills training [7,8]. In adulthood, autistic individuals need support for advanced education or vocational success [9,10].

Although specific practice guidelines are available [11,12], autistic individuals usually experience inadequate access to healthcare, poor diagnostic and post-diagnostic support, and inefficiencies in intervention, residential, education, employment, financial, and social support services [13,14,15,16,17,18,19,20,21]. The reasons for users’ perceptions and experiences of inefficiencies may be due to the lack of a single care system that delivers services across all domains or life stages and effective coordination between different health/social care providers and services. The care of autistic individuals generally requires multiple sectors and providers in different stages of life, and healthcare systems should recognize the need for integration across systems to support the needs of autistic individuals and their families [3]. The integration, coordination, and transition of care are significant lifelong challenges for autistic individuals, their families, and service providers/professionals. In 2022, the Lancet Commission on the future of care and clinical research in autism [3] described a personalized health model of assessment and intervention for individuals with autism to account for their varying needs. The treatment delivery system should provide everything, from the least resource-intensive service to more intensive or specialist-delivered treatments, if necessary, and include multiple dimensions in identification, assessment, and treatment. The assessments should focus on information relevant to treatment planning in collaboration with families. The follow-up care should consider mutual goals set out by the clinician and the autistic individuals and their families to monitor progress and ongoing service needs. In addition, to emphasizing family and individual priorities, personalized stepped care should highlight ways to use community resources within the context of the family and community [3].

In 2016, the Italian Ministry of Health funded and established, at the National Insitute of Health (Istituto Superiore di Sanità), the ‘National Observatory of ASD’ (OssNA), “https://osservatorionazionaleautismo.iss.it/ (accessed on 22 March 2023)” to promote actions targeted at improving the care of autistic individuals. Since then, the OssNA has supported the Ministry of Health in promoting projects aimed at integrating evidence-based practices into care systems at regional and local levels. Actions included developing educational and health services training, strengthening clinical networks, implementing a computerized infrastructure to facilitate communication among professionals, and supporting local initiatives to improve continuity of care. A national technical committee, including representatives of autistic individuals, set the priorities to be developed and financed.

OssNA has been engaged in the development of an integrated care pathway from the recognition of early signs to support in the management and treatment of ASD. In Italy, the central government (the Ministry of Health) sets and coordinates the national health policies and priorities; whereas, regions and autonomous provinces organize the macrostructure of their health systems and deliver care through the local health units “https://www.commonwealthfund.org/international-health-policy-center/system-features/what-are-major-strategies-ensure-quality-care (accessed on 22 March 2023)”. Currently, the central government has issued principles for organizing an integrated plan of care for autistic individuals “https://osservatorionazionaleautismo.iss.it/linee-di-indirizzo (accessed on 22 March 2023)” and each region has independently implemented services in a differentiated manner. Public health researchers of the OssNA aimed at identifying and promoting, at the national and regional levels, the effective components of an integrated care pathway, including the interaction between the health system and other services such as education and social services.

The present project supported this aim by reviewing the international literature on the integrated care pathway for ASD and exploring the European legislative actions on care continuity and care individualization recommendations.

## 2. Materials and Methods

We investigated the integrated care pathways for ASD across the lifespan following two steps: (1) the review of the international literature using the bibliographic databases PubMed and PsycInfo; (2) the review of the European legislative actions on care continuity and care individualization recommendations from childhood to adulthood for autistic people by consulting the Autism Spectrum Disorders in Europe (ASDEU) project partners.

### 2.1. International Literature Review

The review has been elaborated following the Preferred Reporting Items for Systematic reviews and Meta-Analyses (PRISMA) guidelines [22]. The protocol for this review was registered with PROSPERO database: CRD42021233635.

We performed search strategies of articles indexed from inception to 16 September 2022 using the bibliographic databases PubMed and PsycInfo. The highly sensitive search strategies were developed using a combination of MeSH (Medical Subject Headings) and terms to capture the available literature on the topic.

The search strategies were framed into the PICO format (Population, Intervention, Comparison, Outcome) and focused on Population and Intervention domains (See Table 1). No language and temporal restrictions have been applied.

The process of identification, screening, and inclusion of studies is presented in the PRISMA Flow Diagram (Figure 1; [22]). The 6949 publications detected through the search strategy were collected in the Systematic Review Rayyan QCRI application [23]. Four independent authors screened titles and abstracts and excluded the 6264 records papers that did not describe integrated care pathways for individuals diagnosed with ASD. The same authors evaluated the full text of the 606-remaining works. The works were included if the following criteria were met:Description of a structured and stepped multidisciplinary plan of care with timeframes or criteria-based progression from referral to care management and post-diagnosis support.Population with a diagnosis of ASD, Asperger Syndrome, Child Development Disorders or Pervasive Developmental Disorder performed using the Diagnostic and Statistical Manual of Mental Disorders-IV (DSM-IV) [24], DSM-IV-TR [25], DSM5 [1], International Statistical Classification of Diseases and Related Health Problems-10 [26], or scores above a clinical threshold.

Disagreements were resolved by a consensus meeting. Data were extracted for the formal narrative synthesis, including general information about the study and relevant findings. Included articles were assessed through the Critical Appraisal Skills Programme Qualitative Study Checklist (Appendix A).

### 2.2. European Policies and Legislations Search Strategy

We requested, from the partners of the ASDEU project “https://osservatorionazionaleautismo.iss.it/attività-di-ricerca-iss2 (accessed on 22 March 2023)” the national guidelines and legislation on continuity and individualization of care from childhood to adulthood for autistic persons implemented in their countries.

The ASDEU members are experts across 14 European countries (Austria, Belgium, Bulgaria, Denmark, Finland, France, Iceland, Ireland, Italy, Portugal, Poland, Romania, Spain, and the United Kingdom). The ASDEU project, based on the financial support of the Directorate-General of Health and Consumers of the European Commission (DG-SANCO), studied the prevalence of ASD in 12 European countries, analyzed the economic and social costs of ASD, and improved the understanding of diagnosis, interventions, care, and support for autistic children and adults, their families, and professionals. The National Institute of Health’s (Istituto Superiore di Sanità) researchers sent an e-mail to all ASDEU partners clarifying the purpose of the request and asking for documents related to their national guidelines and legislations for ASD. All documents were evaluated for specific components useful in defining an integrated care pathway for ASD.

## 3. Results

### 3.1. International Literature Review

Five works reported a multidisciplinary plan of care with criteria-based progression from referral to managing the condition post-diagnosis [27,28,29,30,31] (See Table 2 and Table 3, Section 3.1.1.). Among studies assessed in full text, a relatively large group of works (N = 606) described organizational aspects, procedures applied to translate recommendations or evidence into local structures, and ‘inventory of actions’ adopted to improve the healthcare pathway. Despite these manuscripts’ lack to meet inclusion criteria, they reported insights and supported actions that may contribute to the definition of an integrated care pathway. We summarized the leading thematic area and provided some examples of interventions described in the studies (See Section 3.1.2.).

#### 3.1.1. Multidisciplinary Plans of Care from Referral to Post-Diagnosis

According to their viewpoint, Green and colleagues [27] explained that an integrated care pathway in autism requires a combination of early detection, pre-diagnostic, and post-diagnostic family-focused, parent mediated interventions, and later step-up and step-down support [27]. They described in detail a proactive and developmentally phased care pathway for young autistic children (up to age 11 years), and considered that a similar pathway could be prepared for autistic adolescents and adults. The application of new digital health technologies (i.e., smartphone apps, integrated digital records, digital navigators, managed online communities, virtual reality) could be a key part of the healthcare system to support and enable the early autism care pathway.

Kong and colleagues [28] described the “Systematic Network of Autism Primary Care Services (SYNAPSE)”, a coordinated healthcare grounded in primary care. The network consisted of ASD-trained primary care (PC) providers and aimed at implementing ASD-specific action plans. These latter included protocols and checklists that empower the PC provider’s role as “synapses”, integrating multidisciplinary function teams and energizing connections among actors of ASD management. Protocols included the early screening of high-risk children, transition to adulthood, home evaluation, inpatient navigation, parenting support, and development of patient registries to track the progress of care over time for a detailed picture of the Network (see details in [28]). The authors emphasized this approach as a proactive care plan for preventing, diagnosing, and treating comorbidities (within the capacity of the PC team). The PC team and specialists can also develop a workflow to promote bidirectional collaboration with services in the community (e.g., Individual Transition Plans in school) and the medical system.

Conversely, Fueyo and colleagues [29] considered the implementation of a health home model in a specialty behavioral health setting rather than in a primary care setting. Care consisted of a diagnostic evaluation by a multidisciplinary team, including case conceptualization, the definition of the plan of care, periodic reassessments for evolving needs, and treatment monitoring. Highly trained life-care managers would function with a broad understanding of the needs of patients and families and would be supervised by masters-level clinicians from the multidisciplinary team. The health home team would work closely with school districts, vocational training agencies, adult transition services, and sheltered workshops. The expertise of the multidisciplinary team is needed to address impairments in functioning, behaviors, and comorbidity. The ASD health home model would also include the use of an electronic patient registry and the integration with general medical services to provide comprehensive care.

With differences in aims and approaches, two studies based on the Action Plans for ASD carried out by Rutherford and colleagues [30,31] were also included in the review. The first study [30] examined the effect of a multi-level modeling approach on overall wait times for diagnostic assessment of ASD. The results consist of significant reductions in waiting times and an increased proportion of girls identified. The waiting time was reduced from referral to diagnosis shared in a children’s service jointly by Child and Adolescent Mental Health Services, Choice and Partnership approach and Speech, and Language Therapy. The study took place in one of the 14 health boards across Scotland with a population of 850,000 and representing 16% of the Scottish population. The study provided the local action plan and included seven key areas for improvement: devising and implementing a care pathway; improving information gathered and shared at the pre-referral stage; reducing the wait for the first appointment; reducing the assessment duration at different stages in the pathway; providing effective post-diagnostic support; improvements in quality and adherence to ASD clinical guidelines; and evaluation to support ongoing review. A range of factors could contribute to more effective use of clinical time: better referral guidance, use of the abbreviated pathway, and triage, together with a planned approach to training and mentoring within all members of diagnosing teams and not just a few expert practitioners.

The second study [31] implemented a 12-month change program, together with ASD-specific approaches, to significantly reduce the waiting time for autism diagnosis in adult health services. Rutherford and colleagues [31] explored if waiting time for ASD diagnosis could be reduced by the pathways and documentation designed to facilitate the reduction in waiting times for diagnostic assessment [32], and the implementation of the “Flightgate” practice development interventions used within mental health services [33]. Participating services were identified from the Diagnosing Services National List which acted as the sampling frame of all services providing assessment and diagnosis of autistic adults in Scotland. They examined waiting times for diagnostic assessment of ASD in 11 adult services, prior to and following the implementation of a 12-month change program. The results were statistically significant and provided a promising framework for service improvement to reduce the wait for diagnostic assessment of ASD in adults across the range of spectrum presentations.

#### 3.1.2. Actions Aimed at Improving Healthcare Procedures

##### Early Recognition, Screening, Diagnosis, and Intervention for Autistic People

The international literature reported several organizational actions aimed at facilitating timely diagnostic evaluation and intervention programs for children with ASD. Levy and colleagues [34] systematically reviewed the evidence for universal screening of children for ASD in primary care (PC) and found evidence for moderate to high positive predictive values for ASD screening tools to detect children 16 to 48 months of age in PC and PC-like settings. However, efficient models of care are needed to provide rapid and accurate diagnosis and intervention for children with concern for ASD. Clinical and policy groups will likely continue to guide and decide about screening and recognition practices.

McNally and colleagues [35] recently experimented with the Early Autism Evaluation (EAE) Hub system, a statewide initiative for ASD screening and diagnosis in the primary care setting. The EAE Hub system provided technical assistance for the development of screening in the primary care setting, community outreach, and training of primary care clinicians in ASD assessment. Through targeted delivery of education, outreach, and intensive practice-based training, large numbers of young children at risk for ASD can be identified, referred to, and evaluated in the local primary care setting.

Crowley, O’Connell, and M. Gervin [36] emphasized the importance of having professional teams within community mental health services who are ASD-trained and supported to address the “core difficulties” of ASD. The professional team provides a ‘hub’ for developing expertise in ASD without intellectual disability, serves inclusive community health organizations, and advises other adult mental health teams on service development and training. This model consists of a planned path from needs assessment to assignment to a group or individual intervention, with the same staff involved in the assessment, intervention, and follow-up stages of treatment [36].

Ibáñez and colleagues [37] examined a healthcare delivery model to improve both early ASD detection and treatment before the age of three. The model declared three components: (1) universal use of ASD screening by PC providers at 18-month well-child visits; (2) immediate referral of positive screens to a community-based early intervention program; (3) provision of an inexpensive, evidence-based ASD-specialized treatment by early intervention providers. This is the first model working simultaneously with two service delivery systems to improve early detection and treatment for ASD. The approach successfully attained buy-in from PC and early intervention providers and built and maintained partnerships with providers.

The South Carolina Act Early Team (SCAET), described by Rotholz and colleagues [38], intended to improve early identification and intervention for young children at risk for ASD through a two-tiered screening process and collaboration among state agencies, universities, health care systems, private organizations, and families. Improvements in early identification and intervention are feasible through collaborative policy change [38].

William-Ayra and colleagues [39] developed the ARENA model, an innovative and multidisciplinary model to improve access to diagnostic evaluations for children younger than three years with concerns for ASD. The model consisted of a care pathway to reduce waiting lists for initial visits and the time needed for final diagnosis by collapsing the number and complexity of visit types, and conducting ongoing measurement and management of supply and demand.

##### Engagement with Services and Shared Plan of Care

The clinical complexity of an individual with ASD requires coordinated interventions. McAllister and colleagues [40] developed an effective and feasible care coordination model, with a Shared Plan of Care (SPoC) used as a family-centered and team-based guide. Team members included clinicians (developmental pediatricians, internal medicine-pediatrics physicians, and clinical psychologists), coordinators (registered nurses, clinical social workers, family leaders), researchers, senior family leaders, and administrators. Five phases of care coordination activities were established: (1) family outreach and engagement; (2) family and team pre-visit work; (3) population care and teamwork; (4) planned care visit; (5) ongoing coordination activities and community transfer. The results showed that a family-centered, team-based, care coordination intervention with a SPoC approach produces outcomes significant to family careers and children’s health and development.

Shepley and colleagues [41] described a brief family-centered service provision model designed in collaboration with healthcare providers where professionals deliver indirect services to children, and carers are the leading actors of the service provision. The program is based on several follow-ups that ensure performance-based feedback. The steps of this model are the following: Professionals and carers define a goal for services, and they collaboratively perform a functional analysis on the child’s severe behavior. Then, professionals establish a function-based treatment including the goal and the carers’ resources and preference. Carers are trained and perform treatment and data collection. Finally, professionals receive feedback on the feasibility of the treatment and adjust it.

A recent systematic review and meta-synthesis [42] explored autistic adults’ mental health service experience, triangulating the perspectives of autistic adults, clinicians, and parents. The collection of qualitative studies synthesized that autistic adults’ experience in current service systems is predominantly negative, with autistic adults facing several barriers when accessing and engaging with support for mental health difficulties. Empowering autistic adults, including them as active participants in the treatment process and facilitating self-care, are necessary steps for effective and inclusive mental health care pathways. Child-focused and adult-orientated healthcare services should work with autistic individuals and their carers to facilitate successful engagement with services and enable adults to manage their mental health needs [43,44].

##### Transition from Adolescence to Adulthood

The importance of the transition processes is recognized by clinicians, researchers, and policymakers [45,46], and the National Institute for Health and Care Excellence published specific guidelines to provide practical recommendations and offer a model for improving the quality of health care transitions [3]. Child and adult mental health services are two separate systems with unique funding structures, services, care philosophies, and target populations. The transition from child to adult services is often fragmented, and there is frequently a loss of diagnosis or continuity of care for autistic individuals.

A study analyzing the Italian Administrative Healthcare Database, from 2010 to 2017, observed a sharp decrease in the percentage of patients receiving a neuropsychiatric consultation after the age of 18, and in the utilization of rehabilitation services with age [47]. Autistic individuals changed clinical referral services as they aged, moving from neuropsychiatric and rehabilitation services to psychiatric and community services as they transitioned from developmental to adult health services.

According to the recent literature, key elements associated with favorable transition outcomes are the implementation of individual educational planning projects and peer-mediated interventions, the promotion of good practices supporting autonomy and transition, and the involvement of parents and teachers in the construction of the person with ASD’s future [48]. Hatfield and colleagues [49] proposed an online transition planning program supporting autistic adolescents to prepare for leaving school and successfully exploring the feasibility and viability of the schools’ programs. White and colleagues [50] and Jonsson and colleagues [51] developed two programs to address the psychosocial and transition-related needs of emerging autistic adults and to empower young adults with ASD to make progress within significant life domains (i.e., work, education, finance, housing/household management, health, leisure/participation in society, and relationships/social network). Autistic youth enrolled in both programs were better prepared to transition out of high school, and these gains were sustained mainly after the programs’ completion.

##### Coordination between Organizations to Facilitate Education and Employment

Autistic students benefit from carefully planned transition, appropriate accommodations, and support to be successful academically and socially. Recently, Accardo and colleagues [52] investigated the accommodations and support services preferred by college students with ASD using sequential mixed methods non-experimental survey and semi-structured follow-up interviews. Students with ASD reported using both academic and non-academic supports with frequency (e.g., extended time on exams, transition program), academic supports in line with other populations with disability, and non-academic supports connecting them one-to-one with a faculty member or coach as preferred (e.g., academic coach, counselor, faculty mentor). Findings suggest a need for university disability service centers, counseling services, and faculty to work together to develop systematic support systems for college students with ASD.

Sung and colleagues [53] explored the feasibility, and efficacy of an employment-related social skills intervention for young adults with autism. They found that manualized, empirically based programs such as Assistive Soft Skills and Employment Trainings promote improvement in social functioning, self-confidence, engagement, and adherence to training, as well as broader impacts, including improvement in sense of belonging and greater employability in this traditionally underserved population. Analogously, Integrated Employment Success Tool successfully improved employers’ self-efficacy in modifying the workplace for individuals on the autism spectrum [54].

##### Health Care Services, Multidisciplinary Approach, and Inpatient Settings

The literature also reported examples of care pathways or actions implemented in inpatient settings. Youth with ASD are admitted to psychiatric units at high rates since ASD specialized units are scarce, even though they have proven to be effective. Kuriakose and colleagues [55] experimented with the ASD Care Pathway (ASD-CP), a scalable approach for improving care in general psychiatric units through staff training and autism-specific intervention strategies. The training with a specific modular staff training, toolkit, and prescribed practice improved the healthcare performances, in terms of decreasing hospital staying and decreasing crisis interventions.

Care pathways should also create challenges for delivering high-quality, patient-centered medical care for autistic individuals. Overall, the findings of the studies [56,57,58,59] successfully revealed strategies to optimize the care of autistic individuals in medical settings: pre-visit planning, anticipating, and reducing sources of distress, facilitating a patient- and family-centered multidisciplinary approach, employing environmental interventions, and using psychopharmacologic treatments. A growing body of literature is devoted to the management of specific needs such as dental or pre-intervention care [60,61,62]. Haydon and colleagues [63] provided reasonable adjustment in any healthcare service to improve the challenges of autistic people. The authors suggested that the adjustments should be identified with an autistic person or a person who may be autistic and recorded on electronic health records so future professionals can be informed. They also suggested that the autistic person should have a health passport with the list of preferences for communications and issues that could cause distress. Finally, among other things, the authors provide reasonable adjustments for effective communication, and overcoming environmental barriers [63].

##### Digital Platforms for the Management of Care Pathways

Several studies suggested that computer automation and electronic health records improved care pathway performances when integrated with the clinical workflow (i.e., screening in primary care) [64,65].

Nicolaidis and colleagues [66] used a community-based participatory research approach to develop and evaluate the AASPIRE Healthcare toolkit, allowing patients to create a personalized accommodation report on general healthcare and autism-related information. Patients and providers involved in the study indicated that the tools positively impacted healthcare interactions, self-efficacy, and patient–provider communication.

Dreiling and colleagues [67] implemented a pilot mental health version of Project ECHO (Extension for Community Healthcare Outcomes) Autism model, by a tele-mentoring platform, aimed at increasing the mental health provider’s knowledge, self-efficacy, and problem-solving about ASD. Analysis of pre- and post-measures revealed improvements in terms of satisfaction about the experience and increase in mental health knowledge and competence. The use of a digital platform to create a connection with the primary health care services and experts may be useful to expand mental health service options for autistic individuals by a sustainable “knowledge network” for the mental health providers and for the trainers too.

### 3.2. European Policies and Legislations

The ASDEU members provided regulations, legislations, policies, and/or national strategies for ten countries (Bulgaria, France, Iceland, Italy, Poland, Portugal, Romania, Spain, United Kingdom, Ireland) and three Spanish autonomous communities (Catalonia, Castile and Leon, Castile de la Mancha). The list of documents sent by the ASDEU members can be consulted in Appendix A. Countries reported autism-specific strategies, guidelines to provide a framework for medical care for autistic people, or legislations for education and employability not specific for autistic people. Several documents called for a collaboration between public bodies in the health, education, and social welfare sectors.

The documents provided contained or cited principles and indications to support the inclusiveness of autistic individuals and to facilitate access to the services. Further, various actions described in the documents aimed at implementing social support, recommendations for early diagnosis, integrated and individualized treatments, and employment support for adults. Clinical recommendations or safeguarding rights for adults were also cited.

## 4. Discussion

The present project aimed at identifying existing care pathways from referral to managing the condition post-diagnosis for autistic people. We reviewed the international literature and collected the European legislative actions on care continuity and care individualization recommendations from childhood to adulthood for autistic people.

Several “building bricks” of care pathway emerged from studies that attempted to develop and build an integrated system considering clinical, living, and educational settings for autistic children and adults [27,28,29,30,31]. Further, a relatively large group of studies described insights and support actions that may contribute to identifying the “building bricks” component of an integrated care pathway for autistic people. These studies described procedures aimed at improving early recognition, screening, and diagnosis of children before the age of three, advancing interventions, reducing waiting lists for initial visits, and involving team members and carers in the care planning. One topic debated was the transition from adolescence to adulthood, which often involves several challenges for people with autism and their caregivers, complicated by the transition from developmental to adult health services in many countries. A beneficial transition outcome is determined by the involvement of peers, parents, and teachers in the construction of plans for the autistic adolescent [49]. It has been observed that medical care and hospital experiences are more positive if professionals received autism-specific training for intervention strategies [55]. Finally, the care pathway performances can be improved by the digitalization and automatization of the health record collection [27].

The European legislative actions on care continuity and care individualization recommendations from childhood to adulthood for autistic people were collected through the ASDEU project members. These documents are oriented toward identifying objectives concerning the integration of autistic people in society, at various ages, and the protection of their rights. Overall, the documents aimed at protecting and empowering autistic people, but details on integrated care pathways aided by integrated care for autistic individuals were not systematically provided. Despite the heterogeneity of strategies and actions implemented in different countries, the need for coordination of care was repeatedly found among documents. The value of defining and activating shared pathways between the territorial and regional realities is evident, regardless of the different organizational structures, which guarantee quality and homogeneity and the use of evidence-based interventions for the assumption of responsibility.

A care pathway has been defined as “a system designed to improve the overall quality of healthcare by standardizing the care process and promoting organized, efficient service user care based on best evidence to optimize service user outcomes” [68]. Within this framework, the organization of services should include the development of a “local autism multi-agency strategy group”, with the capabilities of managing the complex necessities of the autistic people by hearing the voices of stakeholders, autistic persons, and their families. The assessment, management, and coordination of care for autistic children and young people should be provided with professionalism from health, mental health, learning disability, education, and social care services [68]. Further, communication and coordination between multiple organizations/disciplines should be promoted. Professional teams should formulate not only the diagnosis but also define the individualized assistance projects, assess their progress, and carry out advisory activities in different contexts of life (e.g., educational, work) of the autistic person. The multidisciplinary team must consider the individual characteristics, expectations, and preferences of the autistic person, together with the context in which the autistic person and their family live, with the aim of improving their quality of life. Autistic individuals benefit from approachable staff specialized in understanding autism, as well as flexible opportunities for social interaction and approachable communication options that could facilitate access to existing social services [69]. Family engagement, participation, and co-design of care should be also considered as “building bricks” of the integrated care pathway.

The Parents’ Associations have been complaining for years about the lack of personnel specially trained in the care and education of autistic individuals. The professional training must be aimed at implementing evidence-based instruments and procedures to assess and monitor development trajectories, as well as promote the delivery of interventions and support based on the best available scientific evidence. This should be conducted to facilitate the construction and/or participation in individual life projects and support programs aimed at improving the quality of life of autistic people and their families. Training will include methods for monitoring outcomes through appropriate indicators such as quality of life of the autistic person and their family. Providers should also deliver parental training and psychological support to families and carers. Clinical pathways link evidence to practice reducing variation, improving quality of care, and maximizing patients’ outcomes [70]. Healthcare managers rely on clinical pathways to improve health organizations’ efficiency in providing diagnoses and interventions [71]. Autistic individuals and their carers should be supported by comprehensive, integrated, and responsive health and social care services, where the clinical pathway is well defined according to their needs and integrated with all aspects of people’s lives.

The complex and heterogeneous nature of ASD leads to the need for multidisciplinary care and effective coordination between different health/social care providers and services. As reported by Choi et colleagues [72], multiple disciplinary approaches are necessary to resolve complex problems, and multiple disciplinary teams have been found with success in situations such as consensus clinical definitions for complex diseases, and comprehensive health care services and health education.

Autistic people change clinical referral services with age, moving from child psychiatric and rehabilitation services to adult psychiatric and community services. For these reasons, an integrated care pathway ‘model’ should be centered around the concepts of case management, individualization, facilitation, support, continuous training and updating, and quality management. The case-management of autistic persons should consider, in the diagnostic time, accurate and culturally sensitive screening approaches and special attention to the co-occurring conditions, frequent in autistic persons. Decision-making should be shared with the autistic person, where possible, and families, in all phases of their life [73]. Educators and carers should encourage children and young people to express their views and make shared choices. If it is not possible to fully express their vision, it is important to involve people close to them and consider that not all individuals with the same diagnosis have the same needs [74]. In an extraordinary situation, such as pandemic periods, an urgent update of existing policies and guidelines on the accessibility of COVID-19 services to meet the special and different needs of autistic persons and to prevent their exclusion from services [75], also by considering the positive effect of using telehealth in terms of accessibility for adults with ASD, families, and teachers [76], is recommended.

Based on these observations, although diagnostic tools and behavioral/pharmacologic interventions may change over time and based on up-to-date scientific evidence, several challenges in providing high-quality, patient-centered medical care for autistic individuals are critical and have been identified by the analysis of the international literature and European guidelines/legislations. The creation of an integrated care pathway for ASD may be facilitated by the following factors: (1) establishing clinical-diagnostic-functional assessment pathways and monitoring the adherence of health services and professionals to the national guidelines for ASD; (2) re-organizing and providing coordination between health–social–educational services to ensure continuity of care, especially during the delicate phase of transition between child psychiatric services and mental health departments; (3) implementing differentiated intervention and support paths (through the definition of the Individual Rehabilitation Therapeutic Plan) on the basis of the different needs, level of adaptive functioning, and any associated medical or psychiatric co-occurring conditions; (4) promoting educational methodologies defined in a tailor-made way for each individual to activate the maximum possible autonomy and improvement of quality of life, by promoting, for example, employment and social inclusion, and selecting the more appropriate housing solution such as independent living, apartment groups, and co-housing. The local care network should be developed adopting the full spectrum of cited “building bricks”, and adapted in a fluid and coordinated way to the different needs over the course of the autistic person’s life, ensuring particular attention to the transition phases and continuity of care into adulthood. The Lancet Commission [3] has defined “a system of care loosely so as to include the set of health, education, social care, employment, financial, and safety net services, including informal networks or relationships, that families and autistic people potentially have access to in a given community. This definition includes both general systems of health and education and systems, programmes, or benefits targeted at people with disabilities or special needs”. Changing systems of care can improve outcomes for autistic people and their implementation involves legal considerations.

The present review provided insights that must be considered in light of some limitations. First, the literature search may have excluded some relevant documents, as it was limited to English works and considered materials sent by the members of a European project (ASDEU). This may have precluded the understanding of the depth of findings in a more global context. Second, we adopted a strict inclusion criterion for studies, and only a few studies among more than 600 were presented in Section 3.1.2. Third, the present work focused on ASD and may have excluded important studies on care pathways dedicated to mental health conditions. Future research should explore care pathways for children, adolescents, and adults with mental health and neurodevelopmental conditions, expanding the search strategy to the full spectrum of the component of a care pathway and then systematically investigating all the stages of this through further research, conferring a greater degree of detail.

## 5. Conclusions

The present work reviewed the available evidence on ASD care pathways and services and guidelines for the management of children and autistic adults and interrogated the ASDEU members on European policies and legislation. It experimented with solutions to ensure health, improve living and educational conditions, and inclusion in society and the workplace for autistic people. The main aim of the integrated care pathway is to enhance the quality of care by improving individual outcomes, promoting safety, increasing people’s and carers’ satisfaction, and optimizing the use of resources. More research is needed to promote the development of clinical pathways that integrate the recommendations of the international consensus documents with the specific needs of people with autism. The needs and preferences of autistic people should always guide the choice of the specific care pathway, and their families should always be involved in decisions about their health and care.

## Figures and Tables

**Figure 1 ijms-24-06222-f001:**
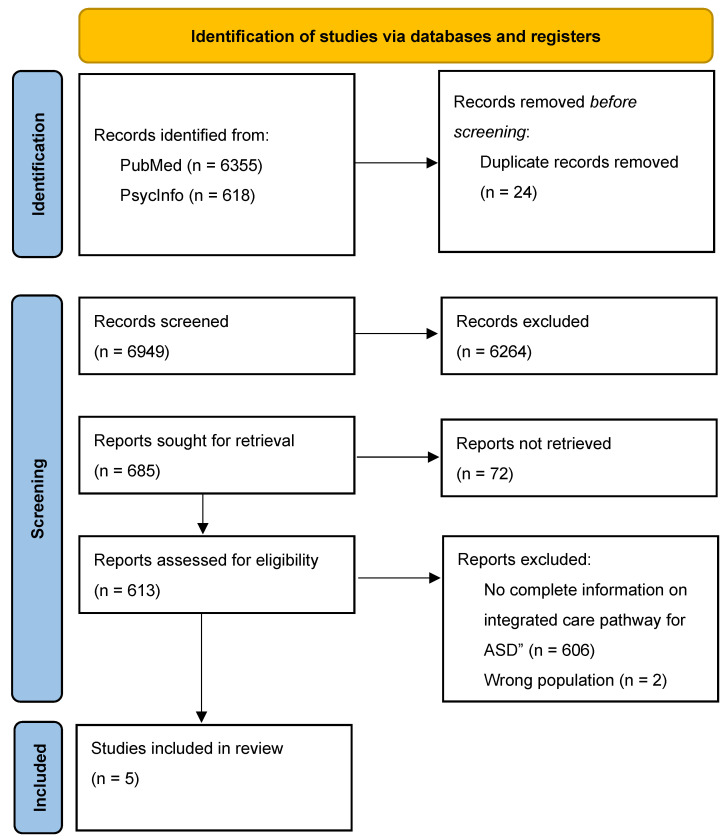
PRISMA 2020 Flow Diagram presenting study screening and selection process From: Page et al. [22].

**Table 1 ijms-24-06222-t001:** The search domains for the database PubMed.

Domain	Search Strategy
Population	Autistic Disorder [Mesh] OR Autism Spectrum Disorder [MeSH] OR Child Development Disorders, Pervasive [MeSH] OR Asperger Syndrome [MeSH] OR “pervasive developmental disorder”
Intervention	“Integrated carepathways” OR “Integrated care pathway” OR “Care coordination” OR “Care coordinator” OR “Care coordinators” OR “Care management” OR “Care maps” OR “Care map” OR “Care model” OR “Care models” OR “Care paths” OR “Care path” OR “Care pathways” OR “Care pathway” OR “Case management” OR “Case management plans” OR “Case management plan” OR Case manager [MeSH] OR Outcome and Process Assessment OR Outcome and Process Assessment (Health Care) OR “Clinical care pathways” OR “Clinical care pathway” OR Continuity of patient care [MeSH] OR Critical pathways [MeSH] OR “Empowerment” OR Delivery of Health Care, Integrated [MeSH] OR Health Services [MeSH] OR “Health care organization” OR “Health care services” OR “Health care service” OR “Lean approach” OR “Lean approaches” OR “Lean management” OR “Patient care” OR Patient care management [MeSH] OR “Patient centred model” OR “Patient centered model” OR “Patient centred models” OR “Patient centered models” OR “Patient-care” OR “Patient-centered care” OR “Patient-centred care” OR “Social care organization” OR “Social care services” OR “Social care service” OR “Care Transitions” OR Transition, Care OR Transitions, Care OR Transition of Care OR Health Care Transition OR Care Transition, Health OR Care Transitions, Health OR Health Care Transitions OR Transition, Health Care OR Transitions, Health Care OR Transitional care [MeSH] OR Management Service Organizations [MeSH] OR “Diagnostic-therapeutic-assistance pathway” OR “Employment” OR Employment, Supported [Mesh] OR Residence Characteristics [MeSH] OR “Housing” OR Continuing Care Retirement Centers OR Life Care Centers, Retirement OR “Residential housing” OR “Supported living” OR “Transitional models” OR “Supported residence” OR “Supported residences” OR Group home [MeSH] OR Residential Facilities [MeSH] OR Intermediate Care Facilities [MeSH])

**Table 2 ijms-24-06222-t002:** Elements of plans of care from recognition to post-diagnosis.

Studies	Plans of Care from Referral to Post-Diagnosis
Green 2022 [27]	Implementation of the integrated and proactive care pathway through immediate and medium-term actions for the following steps: early detection and care, at diagnosis, immediate care, longer-term care; digital health technologies for practitioners and parents; family-focused interventions; step-up care solutions for co-occurring in autism.
Kong 2020 [28]	Networks of ASD-trained primary care providers: implementing ASD specific action plans; integrating multidisciplinary function teams and energizing connections for ASD management; promoting collaboration between the social and health system; proactive care planning for preventing, diagnosing, and treating comorbidities.
Fueyo 2015 [29]	Specialty behavioral health professionals: multidisciplinary team for diagnostic evaluation, case conceptualization, the definition of the initial plan of care, periodic reassessments for evolving needs, and treatment monitoring; trained life-care managers supervised by masters-level clinicians from the multidisciplinary team; collaboration with school districts, vocational training agencies, adult transition services, and sheltered workshops.
Rutherford 2018 [30]	Implementation of actions for children with ASD in key areas: devising and implementing a care pathway; improving information at the pre-referral stage; reducing the first visit waiting time; reducing the assessment duration at different stages; providing effective post-diagnostic support; improving quality and adherence to ASD clinical guidelines; evaluating results to support ongoing review.
Rutherford 2018 [31]	Implementation of actions for adults with ASD through pathways and documentation designed to facilitate the reduction in waiting times for diagnostic assessment; “Flightgate” practice * used within mental health services.

* See the text for details.

**Table 3 ijms-24-06222-t003:** Pathway of care of included studies.

	Phases	Service/Pattern	Coordination	Facilitation	Training	Note
[27]	Pre diagnosis: identify neurodivergent development and pre-diagnosis care. Around diagnosis: support family understanding and adjustment. Post diagnosis: family focused intervention. Long term support: family or carer management/CM.	CM. Step-up and step-down care during transition points or to react to co-occurring conditions.	CM sustains family self-care, resilience. CM: interface between family support and multi-agency collaborative care (health, social care, education).	Digital health technologies. Digital care navigators.	Effective intervention. ASD monitoring with (HV protocols and education assessments).	Large datasets; learning health system. Co-production in service design. Collaborative care.
[28]	ASD early detection. Comprehensive evaluation package. Management of psychiatric and medical co-occurring conditions. Comprehensive functional evaluation and care.	ASD-trained PCPs: coordinate care linked with ASD specialist networks, standardized AP co-developed by individuals, families, providers.	Collaboration with special education, community, adult transitions. Bidirectional collaboration with services in community and medical system.	Accessible workforce training connecting/outreach and IT platform.	ASD-specific PCPs training. Evidence-based care and protocols.	Co-production of care. Organizing care, family/multidisciplinary/interprofessional discussions.
[29]	Diagnostic evaluation by multidisciplinary team; case conceptualization and initial care plan. Periodic reassessment; clinical monitoring. Comprehensive care.	Regional comprehensive treatment center; ASD team in behavioral health setting. HM integration with PC/GM services.	HM team works closely with school districts, vocational training agencies, adult transition services, sheltered workshops.	Electronic health records. Monitoring survey instruments.	Highly trained life-care managers. Life-care manual.	Patients, families, and team work collaboratively. HM.
[30]	AP implementation for identification, referral for specialist assessment; diagnostic assessment; post diagnostic support; quality; adherence to CG.	CP/CSLPT assessment; CAHMS, Choice and Partnership Appointment or CAHMS caseworker assessment; Specialist assessment.	-	-	Training according to levels of skill required by staff.	AP from [32] and guidelines *
[31]	AP implementation for identification, diagnostic process, nonattendance rates inappropriate referrals, efficient working and communication, effectiveness of care pathways.	Health Services with mixed ASD service history.	-	-	Training according to needs in service.	Specific local targets and action plan based on [32] and guidelines *

AP: action plan; ASD: Autism spectrum disorder; CAMHS: Child and adolescent Mental Health Services; CG: clinical guidelines; CM: case management; CP: Community Pediatricians; CSPLT: Community Speech and Language Therapy; HM: health home; HV health visitor ID: intellectual disabilities; GM: general medical; PCP: primary care provider; PC: primary care; * For details on guidelines see [30,31].

## Data Availability

Not applicable.

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
