# Peer review of "Building Bricks of Integrated Care Pathway for Autism Spectrum Disorder: A Systematic Review"

_ijms, 2023, doi:10.3390/ijms24076222_

Round 1

Reviewer 1 Report

Title: Integrated care pathway for autism spectrum disorder: A systematic review and Overview…
First Author: Francesca Fulceri
Manuscript No.:
ijms-2216456

This is an interesting, well-written and well-structured article with important messages.
Moreover, the paper addresses another important (and popular) theme: integrated care pathways and the construction thereof. A strong point of the paper is that science (evidence from literature), health policy making and legal aspects come together. A potential weakness is that the reported studies appear rather diverse and heterogeneous, and that it is not easy for readers to capture all the differences.

Below please do find several suggestions that may be helpful to improve the paper.

Title
1.Lines 2-3. “Systematic review” is a somewhat misleading phrase as “systematic review” is the term used for formal meta-analysis with all the usual elements involved, as critical assessment of the quality of the literature, pooled effect sizes, Forest plot, funnel plot, and evaluation of study heterogeneity, to name a few. But that is not what the authors did. I agree that the paper is a review and gives a qualitative overview of the selected literature, but it is not a systematic review in the sense of a meta-analysis. My advise is to change the title.

Abstract
1. Line 3. ‘Overview’ probably should be written with a lower care ‘o’.
2.Line 17. “…in Europe project partners…” should probably be rewritten as “from European project partners” or “in European projects”.
3.Lines 18-20. Very long sentence. Split this sentence into two sentences, probably in the middle of line19.
4.Line 24. Not sure what “accompaniment” means.

Introduction
1. Line 40. What is “… personalized health model…”? Please define this term, and elaborate somewhat on the concept. It ’s probably a crucial term, because it could be at the model underlying the care pathway.

2.Line 93. Please do change the word “interrogated”.

3. Line 109. “…blinded authors..” makes no sense. Perhaps the authors mean “blinded reviewers “…? Please add why it was important that the review was blinded. Perhaps the researchers mean that the reviewers were also the authors of this paper…?

4. Figure 1. The Prisma protocol and flow diagram are always useful and supportive tools when doing literature search (or meta analysis). Two unclarities need to be resolved: 1) Explain what “reports not retrieved” means. Were these reports not accessible, retrievable, available? 2) Explain what “wrong interventions” means, as no specific definitions or (inclusion) criteria were apparently used to define intervention in terms of good or wrong interventions. So readers cannot grasp, nor judge, what was wrong about wrong interventions, and what are considered good interventions.

5. Line 161, section 3.1.1. and further. I found the literature overview quite interesting and insightful. But I also noticed considerable differences between the reported studies e.g., in aims, design, focus and approach. Furthermore, I found the literature overview rather descriptive (i.e. each study is adequately summarized in its own right). My recommendation is to add qualitative analysis such as comparisons, reflections, interpretations and critial appraisal to the summarized literature/studies, regarding several themes that are important for the development of care pathways (in ASD), so that readers can more easily grasp the key characteristics of the papers. The key characteristics of integrated care pathways (partly addressed in the Discussion) may be helpful to structure this. Perhaps the terms used in lines 472-474 may be helpful to achieve this. Specifically, I’m thinking of a few tables that summarize the main characteristics for each study.
The authors did this already in part, as they selected several themes for detailed discussion: see the headings in lines 229, 302, 329 and 353. I feel more of such themes could be added, such as information, communication and coordination between multiple caregivers/organisations/disciplines; the role of regular healthcare/social services/social support; young vs. adult patients with ASD (and implications for role functioning, like school, work/employment, leisure time activities,…); referral/screening/diagnostic process (including genetics for some ASD patient groups…??)

6.Line 372, section 3.2. This reviewer is not a legal expert nor expert in health law, but my overall impression is that this section is described rather concisely, and that this could gain value with somewhat more qualitative analysis.

Discussion
A few thoughts emerged after reading the Discussion. Perhaps the authors are willing to reflect on this...?

1) Given the different European health systems and health legal systems, is it desirable, feasible to achieve one overall European care pathway for all European patients with ASD ?

2) Regardless the answer to 1): Following the literature review, in the authors’ opinions, what are the elements or “building bricks” that should always be part of an integrated care pathway for patients with ASD? Is the answer different for patients with mild and severe ASD?

3) What are the themes or elements in ASD and in the current care pathways that are missing or underdeveloped and need improvement (critical assessment)? Is there evidence form the literature to support this, as well as guide future directions? In what way should the legal system be adapted to accommodate these improvements?

4) According to the authors, what are the recommendations and future directions for health policy setting?

[ end of review]

Author Response

Many thanks for your careful revision and helpful comments. We believe that the manuscript is substantially improved after addressing your comments and suggestions.

Title
Point 1. Lines 2-3. “Systematic review” is a somewhat misleading phrase as “systematic review” is the term used for formal meta-analysis with all the usual elements involved, as critical assessment of the quality of the literature, pooled effect sizes, Forest plot, funnel plot, and evaluation of study heterogeneity, to name a few. But that is not what the authors did. I agree that the paper is a review and gives a qualitative overview of the selected literature, but it is not a systematic review in the sense of a meta-analysis. My advise is to change the title.

Response 1.  We thank the Reviewer for this comment because it made us reflect on aspects that were missing in our review.  In this updated version of the manuscript, we now provided the assessment of the quality of the literature using the 10-question Critical Appraisal Skills Programme Qualitative Study Checklist (lines 137-138; Supplementary material 1) and we added a structured synthesis of results according to reviewer suggestion (Table 3). We did not combine results of the individual studies to produce an overall statistical meta-analysis because the studies’ results were extremely heterogeneous, but our work followed the PRISMA criteria for conducting a systematic review. We have now updated also the PRISMA checklist for systematic review with the risk of bias assessment information. Further, we thank the Reviewer for suggesting the wording “building bricks” which seems to fit for the review findings. We have adopted this term in the Title.

Point 2. Abstract
1. Line 3. ‘Overview’ probably should be written with a lower care ‘o’.
2.Line 17. “…in Europe project partners…” should probably be rewritten as “from European project partners” or “in European projects”.
3.Lines 18-20. Very long sentence. Split this sentence into two sentences, probably in the middle of line19.
4.Line 24. Not sure what “accompaniment” means.

Response 2.

  1. We have corrected the upper case.
  2. We have changed the sentence as follows: “Autism Spectrum Disorders in Europe (ASDEU) project partners”. Now it appears clearer that the sentence refers to the name of European project.
  3. We have shortened the sentence as follows: “The study found that published data mainly focused on specific components of care pathways rather than an integrated and coordinated plan of care and legislative indications. They should be aimed at facilitating access to the services to support and the inclusiveness of autistic individuals.”
  4. We have replaced “accompainiment” with “support”.

Introduction
Point 3. Line 40. What is “… personalized health model…”? Please define this term, and elaborate somewhat on the concept. It ’s probably a crucial term, because it could be at the model underlying the care pathway.

Response 3. We thank the Reviewer for this comment, which allowed us to introduce the relevance of the concept of personalization of care more effectively. According to “The Lancet Commission on the future of care and clinical research in autism”, the heterogeneity of autism requires personalized assessments and interventions. A personalized health approach takes the heterogeneity of autism into account by recognizing that the profile of strengths and needs of each autistic individual and their family should determine the intervention and support priorities, but also that these can change over time. We have introduced this concept more clearly in the Introduction section and we have also more explicitly described the considerations and model proposed by the committee (line 53 -63). In our opinion, this integration enhances the committee's suggestions and allows the reader to be more easily introduced to the content discussed in the article.

Lord C, Charman T, Havdahl A, Carbone P, Anagnostou E, Boyd B, Carr T, de Vries PJ, Dissanayake C, Divan G, Freitag CM, Gotelli MM, Kasari C, Knapp M, Mundy P, Plank A, Scahill L, Servili C, Shattuck P, Simonoff E, Singer AT, Slonims V, Wang PP, Ysrraelit MC, Jellett R, Pickles A, Cusack J, Howlin P, Szatmari P, Holbrook A, Toolan C, McCauley JB. The Lancet Commission on the future of care and clinical research in autism. Lancet. 2022 Jan 15;399(10321):271-334. doi: 10.1016/S0140-6736(21)01541-5. Epub 2021 Dec 6. Erratum in: Lancet. 2022 Dec 3;400(10367):1926. PMID: 34883054.

Point 4. Line 93. Please do change the word “interrogated”.

Response 4. We have changed “interrogated” with the following sentence: “We searched in PROSPERO (www.crd.york.ac.uk/prospero/) to seek ongoing systematic reviews on care pathways for ASD.”

Point 5. Line 109. “…blinded authors..” makes no sense. Perhaps the authors mean “blinded reviewers “…? Please add why it was important that the review was blinded. Perhaps the researchers mean that the reviewers were also the authors of this paper…?

Response 5. We have replaced “blinded” with “independent”. We have used the term “blinded” because, in the screening process of the title/abstract and full texts, the authors do not know the answers of one and the other. However, we agree that “independent” is clearer. We adopted an independent screening approach to avoid systematic errors, identify, and correct random errors such as careless mistakes.

Point 6. Figure 1. The Prisma protocol and flow diagram are always useful and supportive tools when doing literature search (or meta analysis). Two unclarities need to be resolved: 1) Explain what “reports not retrieved” means. Were these reports not accessible, retrievable, available? 2) Explain what “wrong interventions” means, as no specific definitions or (inclusion) criteria were apparently used to define intervention in terms of good or wrong interventions. So readers cannot grasp, nor judge, what was wrong about wrong interventions, and what are considered good interventions.

Response 6.

1) We adopted the wording “reports not retrieved” according to the official PRISMA. In detail, in Page and colleagues (2021), the PRISMA flowchart is presented with the box “reports not retrieved” which are those articles not available/not accessible in full text.

Page, M. J., McKenzie, J. E., Bossuyt, P. M., Boutron, I., Hoffmann, T. C., Mulrow, C. D., ... & Moher, D. (2021). The PRISMA 2020 statement: an updated guideline for reporting systematic reviews. International journal of surgery, 88, 105906.

2) For “wrong intervention” we wanted to describe the studies that did not provide complete information on the integrated care pathway for ASD. We agree with the Reviewer, and we replaced “wrong intervention” with “No complete information on integrated care pathway for ASD”.  We updated the PRISMA flow chart, accordingly.

The term “intervention” derived from the PICO domains. According to the Reviewer’s concerns, we have reordered the keywords presented in the Table 1 to enable the reader to grasp the meaning of the term more easily. In the previous version of the manuscript, the keywords used for the search strategy were sorted alphabetically, while now the first terms of the search strategy inherent in interventions coincide with the care pathway.

Point 7. Line 161, section 3.1.1. and further. I found the literature overview quite interesting and insightful. But I also noticed considerable differences between the reported studies e.g., in aims, design, focus and approach. Furthermore, I found the literature overview rather descriptive (i.e. each study is adequately summarized in its own right). My recommendation is to add qualitative analysis such as comparisons, reflections, interpretations and critial appraisal to the summarized literature/studies, regarding several themes that are important for the development of care pathways (in ASD), so that readers can more easily grasp the key characteristics of the papers. The key characteristics of integrated care pathways (partly addressed in the Discussion) may be helpful to structure this. Perhaps the terms used in lines 472-474 may be helpful to achieve this. Specifically, I’m thinking of a few tables that summarize the main characteristics for each study.
The authors did this already in part, as they selected several themes for detailed discussion: see the headings in lines 229, 302, 329 and 353. I feel more of such themes could be added, such as information, communication and coordination between multiple caregivers/organisations/disciplines; the role of regular healthcare/social services/social support; young vs. adult patients with ASD (and implications for role functioning, like school, work/employment, leisure time activities,…); referral/screening/diagnostic process (including genetics for some ASD patient groups…??)

Response 7. We thank the Reviewer for this valuable suggestion that gives us the opportunity to improve the Results section. As suggested, we elaborated a new table (see Table 3 pag.8, line 252) describing the main characteristics of the studies included through the terms cited in the Discussion section (i.e., coordination, pattern of pathway, facilitation, etc.). Further, as suggested by the Reviewer, we re-organized e integrated the result section adding various subheadings as “Engagement with services and shared plan of care” and “Coordination between organisations to facilitate education and employment”.  Please note lines 301, 363-382, 384. We added several references, and the order is now different from previous version, accordingly.

Point 8. Line 372, section 3.2. This reviewer is not a legal expert nor expert in health law, but my overall impression is that this section is described rather concisely, and that this could gain value with somewhat more qualitative analysis.

Response 8. The section described the documents sent by the ASDEU members which may not be exhaustive of those available in the European Union. The choice of the sample selection for questioning (ASDE members) was of convenience as collaborators of the Istituto Superiore di Sanità. We tried to perform a census of the available documents; however, they were difficult to translate and to compare. Performing a critical qualitative analysis was beyond our possibility, but this request to researchers with expertise in ASD policy strategies allows us to point out that, in different countries, there is the problem of heterogeneity of care models and therefore a need for future collaboration to overcome these gaps. the outcome "everyone asks for coordination" was the main element that was also useful in understanding that the needs of people with ASD are similar in different systems.

In the new version of the manuscript, we have modified the Methods section describing the procedure adopted to collect these data (lines 144-145). Further, the change in the title in “Building bricks of integrated care pathway for autism spectrum disorder: A systematic review” also downplays the centrality of the request made to the ASDEU collaborators and their contributions.

Discussion
A few thoughts emerged after reading the Discussion. Perhaps the authors are willing to reflect on this...?

Point 9. Given the different European health systems and health legal systems, is it desirable, feasible to achieve one overall European care pathway for all European patients with ASD?

Response 9. We thank the Reviewer for this insight. However, we believe that, given the extreme heterogeneity of the healthcare systems present in Europe, an overall European care pathway is likely difficult to achieve. We also noticed that, even in the Italian context with the Regional and Autonomous Provinces' differences in healthcare management at present, do not make it possible to share a common pathway. The systematic review tells us that there is a need for several “building bricks” as family engagement, participation, co-design, and coordination. These are themes that recur in different geographic and organizational experiences. We have cited this concept in the Discussion section (lines 508-510).

Point 10. Regardless the answer to 1): Following the literature review, in the authors’ opinions, what are the elements or “building bricks” that should always be part of an integrated care pathway for patients with ASD? Is the answer different for patients with mild and severe ASD?

Response 10. We thank the Reviewer for suggesting the wording “building bricks” which seems to fit for the review findings. We have adopted this term in the Title and also in the Discussion section by referring more clearly to the components of relevance that need to be taken into account when constructing a care pathway as family engagement, participation, co-design, and coordination.

This review does not provide the operational details for constructing a differentiated care pathway based on the severity of autism or in occurrence or non-occurrence of associated conditions, e.g., intellectual disability or behavioral disorders. However, we agree with the Reviewer that this is a very important aspect to consider in the care pathway models. In this regard, we have referred more explicitly in the Discussion section to what was proposed by the Lancet Commission by orienting the reader to think about the advantage of a step-down care pathway (lines 571-577).

Point 11. What are the themes or elements in ASD and in the current care pathways that are missing or underdeveloped and need improvement (critical assessment)? Is there evidence form the literature to support this, as well as guide future directions? In what way should the legal system be adapted to accommodate these improvements?

Response 11. The literature tells us that the co-participation and coordination components should guide choices as well as the monitoring of pathways that must be adapted to the needs of those who participate in them (both families and providers). Changes on the legal side are not easy to predict in different contexts but certainly, as indicated by the Lancet Commission, adopting pathways that include all necessary components also involves legal considerations. We cited this point in the Discussion section (lines 576-578).

Point 12. According to the authors, what are the recommendations and future directions for health policy setting?

Response 12.  The aim of this review was to gather experiences and perspectives on the use of effective care pathway models. The literature in this field is limited and particularly complex as well as they are difficult to propose a single operational model that can fit the multiple organizational structures of different areas. However, the review has made it possible to enucleate most of the components that should be considered and suggested to be referred to when constructing care pathways for ASD and their families. We have referred to this more clearly in the Discussion section.

Reviewer 2 Report

This paper systematically reviewed Integrated care pathway for autism spectrum disorder.

The topic fits the scope of International Journal of Molecular Sciences,

In general, the manuscript is well-organized and the references are supportive to the conclusions.

Author Response

Many thanks for your revision and positive comments

Reviewer 3 Report

Submitted systematic review aimed to identify existing care pathways for Autism Spectrum Disorder (ASD) from referral to care management after diagnosis. Review was prepared according to the PRISMA guidelines and registered with PROSPERO system. The review idenitified the need for developing coordinated care and legislative actions allowing better tending to  autistic individuals. It is suitable for publication in IJMS.

Author Response

(The authors gave the same response as above.)

Round 2

Reviewer 1 Report

Title: Building bricks of Integrated care pathway for autism spectrum disorder: A systematic review
First Author: Francesca Fulceri
Manuscript No.:
ijms-2216456 revised version

I have seen the responses to the reviewer suggestions and the changes made to the paper.
First, I want to compliment the authors with the careful and considerate response letter and the revisions in the manuscript.
I still believe the paper addresses several important issues from the ASD as well as integrated care perspective, despite heterogeneity across studies and health systems, and outlines the next steps for research and policy.
Overall, the study provides the international audience a very good and useful overview of the current state of play in care pathways for ASD.

No further suggestions nor comments to add.

[end of review]